# Simulating Two-Sided Job Marketplaces with AI Agents

**AI Agent**

**Silvia Terragni**
Upwork Inc
San Francisco, CA
`silviaterragni@upwork.com`

**Behnaz Nojavanasghari**
Upwork Inc.
San Francisco, CA

**Frank Yang**
Upwork Inc.
San Francisco, CA

**Andrew Rabinovich**
Upwork Inc.
San Francisco, CA

## Abstract

We introduce a simulation framework for studying how artificial intelligence agents behave in economic marketplaces. Unlike traditional computer simulations that use predetermined rules, our approach uses large language models (LLMs) as intelligent agents that can make strategic decisions and adapt their behavior over time. The framework includes reputation systems that track agent performance and detailed logging of decision-making processes.

Through systematic experiments comparing different types of agents, we reveal three key insights: the reasoning capabilities of AI agents create fundamentally different market behaviors, successful marketplaces require compatible decision-making approaches from all participants, and no single performance measure captures market success—instead, there are important trade-offs between transaction volume and match quality. The framework captures both what happens in the market overall and why individual agents make specific decisions.

This work provides a reproducible research platform for investigating how AI decision-making affects economic outcomes in controlled virtual marketplaces. To support reproducibility and further research, we make the complete simulation framework and analysis tools publicly available as open-source software at `https://github.com/upwork/simploy`.

**Disclaimer:** This simulation is illustrative, not prescriptive, and highlights the capabilities and limitations of LLMs in synthetic environments. The framework is not designed to model or evaluate real-world labor platforms or populations, and findings should not be used to draw conclusions about existing economic systems.

## 1 Introduction

Economic research increasingly requires tools for examining complex market behaviors, strategic adaptation, and emergent phenomena under controlled conditions. Traditional agent-based simulations typically rely on predefined utility functions and simplified decision rules (Arthur, 1994, Tesfatsion, 2002). Recent advances in large language models (LLMs), however, make it possible to design agents that approximate decision policies through prompt-conditioned responses and adaptive strategic behavior (Li et al., 2024, Lin et al., 2025, Park et al., 2023, Shinn et al., 2023). This shift opens new opportunities to investigate how cognitive architectures influence economic outcomes.

We introduce a reproducible simulation framework that treats LLMs as bounded policy approximators rather than hand-coded rule followers. The framework is organized around three methodological questions: 1) How can simulation environments be structured to ensure reproducibility while allowing

systematic variation in agent reasoning capabilities? 2) Which architectural features—such as adaptive prompting, reputation mechanisms, and reasoning logs—are most effective for capturing both quantitative outcomes and qualitative decision processes? 3) In what ways can controlled comparative experiments, even in simplified marketplaces, reveal how cognitive architectures shape efficiency, inequality, and strategic adaptation?

To address these questions, the framework integrates adaptive prompting mechanisms, a multi-tier reputation system, and longitudinal behavior tracking into a unified research platform. By leveraging LLMs' ability to produce natural-language reasoning traces, it enables analysis of both aggregate market outcomes and the decision processes underlying them. Methodologically, this design offers several advantages: controlled isolation of economic factors, transparency of agent reasoning, scalability across experimental configurations, and comprehensive data collection spanning behavioral and outcome metrics.

We demonstrate the framework's capabilities in a job-marketplace environment where freelancer and client agents interact through bidding, hiring, and reputation dynamics. These experiments capture distinct economic behaviors—including strategic bidding selectivity, inequality in work distribution, and efficiency trade-offs—arising from differences in agent reasoning capabilities.

Finally, to maintain a clear focus on economic mechanisms, the framework uses anonymous agent identifiers and excludes demographic attributes. It is designed as a research testbed for theoretical inquiry in synthetic environments and is not intended to model or evaluate real-world labor markets or populations.

## 1.1 Contributions

This work makes three primary contributions to economic simulation research:

1. **Simulation Framework:** We provide a controlled environment for LLM-agent interactions that integrates adaptive prompting, a multi-tier reputation system, and longitudinal behavior tracking. This design supports reproducible, systematic variation in agent cognition and market conditions.

2. **Methodological Toolkit:** We contribute an integrated set of analytical components for LLM-based simulations, including comparative experimental design, inequality metrics (e.g., Gini coefficients), explicit reasoning capture through decision logs, reflection-based adaptation, and composite market health indicators. Together, these tools enable interpretable analysis of both strategies and outcomes.

3. **Empirical Insights:** We demonstrate the framework through systematic experiments across five configurations, revealing how cognitive architecture differences drive selectivity alignment effects, fundamental trade-offs between efficiency and equity, and the crucial role of adaptation mechanisms in market outcomes—validating the framework's capacity to uncover economic principles governing synthetic marketplaces.

## 2 Related Work

We review three strands most relevant to our study: (i) agent-based computational economics (ACE), (ii) LLM agents for strategic interaction and market-like environments, and (iii) two-sided market mechanisms and matching. Our focus is on longitudinal strategic adaptation, market mechanism design, and systematic policy parameter investigation through comparative experimental analysis.

### 2.1 Agent-Based Computational Economics with LLM Agents

Classic ACE demonstrates how aggregate regularities can emerge from decentralized interaction among boundedly rational agents (Arthur, 1994, Tesfatsion, 2002). Recent work replaces hand-crafted rules with language-model-driven agents to better capture rich decision processes. The AI Economist couples multi-agent learning with policy design in a simulated economy (Zheng et al., 2022). Building on this direction, Li et al. (2024) introduce *EconAgent*, where LLM agents, equipped with perception and memory, interact in macroeconomic settings. Populations of LLM agents have also been used to study heterogeneous expectation formation in macroeconomics (Lin et al., 2025).

These lines collectively motivate using LLM agents for economic inquiry; however, most results emphasize macro aggregates or single-episode tasks. We instead study repeated, two-sided market interactions with explicit reasoning capture and fine-grained, round-by-round measurements.

## 2.2 LLM Agents for Strategic Interaction and Market-Like Behavior

Systems papers show LLM agents sustaining coherent behavior with memory, planning, and tool use—e.g., social simulations with persistent agents (Park et al., 2023), role-based collaboration (Li et al., 2023), and programmable multi-agent orchestration (Wu et al., 2024). Methodologically, self-improvement techniques such as Reflexion and Self-Refine provide mechanisms for iterative critique and adaptation (Madaan et al., 2023, Shinn et al., 2023), while LLMs can approximate human-like responses in multi-participant settings (Aher et al., 2023).

Closer to markets, agent-based financial simulators populate order books with heterogeneous LLM trading strategies and report behavior aligned with stylized facts under controlled conditions (Gao et al., 2024, Lopez-Lira, 2025). Parallel work probes bargaining and automated negotiation—relevant to bidding and hiring dynamics in two-sided markets—through evaluation platforms and self-play protocols (Bianchi et al., 2024, Fu et al., 2023, Hua et al., 2024, Xia et al., 2024). Our framework draws on these ingredients but targets longitudinal strategic evolution in a labor-style marketplace, with explicit logging of natural-language reasoning to enable mechanism-level analysis beyond outcome aggregates.

## 2.3 Two-Sided Markets and Matching

Empirical and experimental work shows how platform design shapes search, selection, and efficiency in two-sided markets (Fradkin et al., 2021, Pallais, 2014). Reputation is often treated as the central signal mediating frictions, but in practice it interacts with pricing, skill alignment, and past performance. In our framework, we model reputation as one endogenous signal among many, emphasizing how it combines with evolving reasoning processes to influence strategy and inequality over repeated interactions.

**Positioning.** Compared to macro-focused LLM-agent economies (Li et al., 2024, Lin et al., 2025, Zheng et al., 2022) and short-horizon social or negotiation settings (Bianchi et al., 2024, Park et al., 2023), our framework targets a different space: a reproducible, two-sided labor-market simulation with longitudinal tracking. Its design emphasizes comparative variation in agent reasoning, explicit capture of natural-language decision traces, and measurement of how such reasoning shapes both individual strategies and aggregate market patterns. This combination links the interpretability of LLM-driven social simulations with the rigor of traditional agent-based computational economics.

# 3 Framework Architecture and Methodology

This section presents our two primary methodological contributions: (i) the simulation environment architecture and (ii) the analytical toolkit for studying economic behavior.

## 3.1 Simulation Environment

We implement a controlled labor-market simulation where GPT-4o-mini–based freelancer agents compete for jobs posted by client agents. All agent profiles (freelancer personalities, skills, and backgrounds), client company profiles, and job postings are generated by GPT-4o-mini to ensure diverse, realistic market content while maintaining experimental control. The environment integrates adaptive prompting, a multi-tier reputation system, and longitudinal tracking to support reproducible experiments on strategic adaptation.

The market operates through a sequential bidding mechanism: each round consists of job posting, strategic bidding, hiring decisions, reputation updates, and agent reflection. All freelancer agents begin with identical access, zero reputation, and no hand-coded behavioral rules, ensuring that inequality and strategy emerge endogenously. Policy parameters such as population size, bid limits, posting frequency, and reputation requirements are configurable, enabling systematic comparative experiments under varied market conditions.

## 3.2 Agent Architecture

Agents use prompts with anonymized identifiers to avoid demographic bias. Each is constrained to 1–5 bids per round, forcing selective strategies across job categories. To ensure behavioral diversity, agents are assigned distinct GPT-generated personality profiles that influence their decision-making approaches. For example, some agents are "Analytical and methodical, thriving on solving complex problems and enjoying working independently," while others are "Creative and innovative, thriving on feedback and collaboration," or "Meticulous and thorough, enjoying ensuring accuracy in legal matters." Strategic behavior develops through reflection and feedback from the integrated reputation system, without access to market-wide information. Further implementation details appear in Appendix A.1.

## 3.3 Reputation System

The reputation system tracks freelancer outcomes (earnings, jobs completed, category expertise, and tier progression: New $\rightarrow$ Established $\rightarrow$ Expert $\rightarrow$ Elite) and client performance (hire success, spending, budgets). Reputation context is dynamically incorporated into prompts, creating feedback loops between market standing and decision-making. Updates occur after bidding, hiring, and job completion, with longitudinal records enabling analysis of mobility and adaptation.

## 3.4 Analytical Toolkit

The framework collects multi-level data spanning individual actions, reasoning traces, market aggregates, and reputation dynamics. This supports both quantitative analysis (e.g., efficiency, inequality, participation) and qualitative interpretation of decision processes.

**Key indicators.**

- **Fill rate**: fraction of jobs successfully matched. This measures overall market efficiency and indicates how well the platform connects supply and demand.

- **Bid efficiency**: ratio of successful bids to total bids. This captures the quality of bidding strategies, distinguishing between volume-driven and selective approaches to market participation.

- **Participation rate**: proportion of freelancers who submit at least one bid per round (averaged across all rounds). This measures active market engagement and indicates the breadth of freelancer involvement in the bidding process.

- **Freelancer hiring rate**: proportion of freelancers who receive at least one hire. This indicates market inclusivity and whether opportunities are accessible to the broader freelancer population.

- **Bidding selectivity**: proportion of available jobs each agent targets. This reveals strategic decision-making patterns and helps distinguish between opportunistic and focused market behavior.

- **Work distribution inequality**: measured by the Gini coefficient to assess whether work concentrates among a few high-performing freelancers or distributes more evenly across the population

$$G = \frac{2\sum_{i=1}^{n}(i\,x_i)}{n\sum_{i=1}^{n} x_i} - \frac{n+1}{n},$$

where $x_i$ denotes the number of active jobs per freelancer after sorting in nondecreasing order and $n$ is the number of freelancers; $G \in [0,1]$ with 0 indicating perfect equality (all freelancers have the same number of active jobs) and 1 indicating maximum inequality (all jobs concentrated in one freelancer).

- **Market health score**: composite index combining 4–5 factors as an equally-weighted average:

$$H = \frac{1}{n}\sum_{i=1}^{n} f_i, \quad f_i \in [0,1]$$

where the factors are: (1) fill rate $f_1 =$ jobs filled/jobs posted; (2) bid competition balance $f_2$ optimized for 2–4 bids/job via piecewise scaling; (3) low rejection $f_3 = 1 -$ rejection rate; (4) participation rate $f_4 =$ active bidders/total freelancers; and optionally (5) low saturation $f_5 = 1 -$ saturation risk. The score ranges from 0.0 to 1.0, providing a holistic assessment of market functioning. Note that this metric can be misleading when comparing strategic vs. volume-based approaches, as it prioritizes fill rates over match quality.

- **Reputation tiers**: formalized measures of agent standing derived from cumulative earnings and completed work. These capture long-term strategic adaptation and the emergence of hierarchical market structures.

## 3.5 Framework Assumptions and Constraints

The framework makes several assumptions. In terms of market structure, we fix the agent population with no entry or exit, model interactions in discrete rounds rather than continuous time, and apply configurable cooldowns to job postings. On the economic side, we abstract away inflation and transaction costs, assume full visibility of job budgets, and restrict hiring to binary outcomes. Agents make decisions based on structured prompts that incorporate their context, reputation, and past performance. They learn over time through reflection mechanisms that help them adapt their strategies based on market feedback. Finally, technical implementation remains simplified: for example, jobs resolve deterministically and the platform itself does not evolve over the course of a simulation.

## 4 Illustrative Framework Demonstration

We demonstrate the framework's research capabilities through an illustrative comparative study of strategic decision-making differences between LLM-driven agents and random baselines. This example highlights how the framework captures behavioral variation, tracks adaptation patterns, and measures decision-making quality through controlled experiments with statistical validation.

### 4.1 Experimental Design

We evaluate five configurations across two experimental dimensions: (1) agent reasoning capabilities and (2) reflection mechanisms. The primary comparison examines four agent reasoning configurations: **LLM-F + LLM-C** with reflection-enabled reasoning for both freelancers (F) and clients (C), **LLM-F + Rand-C** examining freelancer-side reasoning only, **Rand-F + LLM-C** examining client-side reasoning only, and **Rand-F + Rand-C** with baseline decision-making. Additionally, we conduct an ablation study with **LLM-F + LLM-C (w/o Reflections)** to isolate the impact of strategic adaptation mechanisms.

Each simulation involved 200 freelancer agents and 30 client agents across 100 rounds with identical market parameters: posting cooldowns 2–7 turns, each round a freelancer is randomly shown 5 jobs and can bid only 3 times per round. Each freelancer can have at most 3 active jobs at the time. Random freelancers bid with 5% probability per job and random clients accepted 50% of bids, while LLM agents used contextual reasoning. These random baseline parameters were chosen to reflect realistic market selectivity (not all jobs receive bids, not all bids are accepted) while establishing a clear noise floor that demonstrates when strategic reasoning adds value. The reflection mechanism enables agents to periodically analyze their performance and adapt strategies based on market feedback, with each client and freelancer having a 5% probability of reflection per turn. Each configuration was repeated 20 times.

### 4.2 Comparative Results

The comparative results reveal that no single metric captures marketplace performance; outcomes depend strongly on how freelancer and client selectivity align. Coordinated strategies, whether high or low in selectivity, generate stable outcomes, while mismatched strategies lead to systemic failures.

When both sides are coordinated, different trade-offs emerge. In the high-selectivity setting, LLM-F + LLM-C agents with reflections achieve balanced performance, combining moderate fill rates (69.3%) with strong bid efficiency (25.1%). By contrast, the same configuration without reflections

Table 1: Comparative performance across agent reasoning capabilities and reflection mechanisms (Mean ± 95% CI).

| Metric | LLM-F + LLM-C (w/ Refl.) | LLM-F + LLM-C (w/o Refl.) | LLM-F + Rand-C | Rand-F + LLM-C | Rand-F + Rand-C |
|---|---|---|---|---|---|
| Fill Rate (%) | 69.3 ± 2.2 | 43.5 ± 2.9 | 64.3 ± 2.7 | 38.6 ± 3.4 | **87.7 ± 2.2** |
| Bid Efficiency (%) | 25.1 ± 2.1 | **32.2 ± 1.7** | 20.5 ± 0.4 | 5.5 ± 0.6 | 16.2 ± 0.9 |
| Bids per Job | 2.92 ± 0.47 | **1.35 ± 0.06** | 3.14 ± 0.15 | 7.00 ± 0.11 | 5.54 ± 0.50 |
| Participation Rate (%) | 8.9 ± 1.4 | 4.1 ± 0.2 | 9.7 ± 0.4 | **21.6 ± 0.4** | 17.1 ± 1.5 |
| Freelancer Hiring Rate (%) | 84.6 ± 1.4 | 63.5 ± 3.5 | 85.8 ± 3.2 | 72.7 ± 4.6 | **98.8 ± 0.8** |
| Work Distribution (Gini) | 0.29 ± 0.05 | 0.50 ± 0.03 | 0.26 ± 0.03 | 0.45 ± 0.04 | **0.11 ± 0.06** |

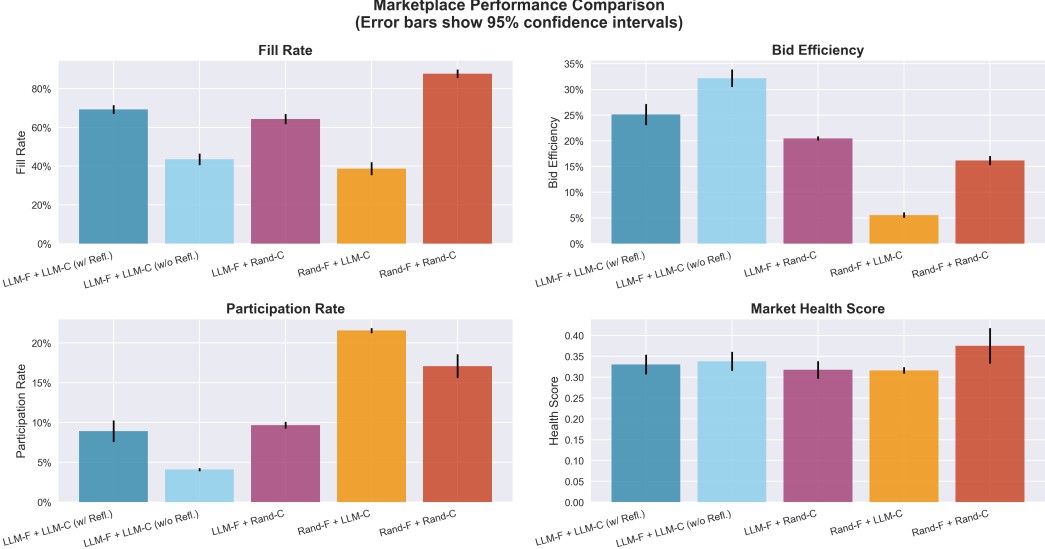

Figure 1: Comparative analysis across five agent configurations including reflection ablation study. (a) Fill rates, (b) Bid efficiency, (c) Participation rates, and (d) Market health scores. Error bars show 95% confidence intervals.

demonstrates the highest individual efficiency (32.2%) through rigid selectivity and narrowly focused bidding (around 1.35 bids/job), but this rigidity reduces overall fill to 43.5%, limiting market integration. At the opposite end of the spectrum, the low-selectivity Rand-F + Rand-C setup maximizes throughput, reaching an 87.7% fill rate and producing the most equitable work distribution (Gini = 0.11). These outcomes illustrate how coordinated selectivity, whether adaptive or indiscriminate, can sustain effective market functioning.

In contrast, hybrid configurations perform poorly because selectivity is misaligned between freelancers and clients. In Rand-F + LLM-C, non-selective freelancers flood the market with indiscriminate bids, but selective clients systematically reject them, resulting in severe inefficiency (38.6% fill, 5.5% efficiency). The reverse configuration, LLM-F + Rand-C, also underperforms: selective freelancers target opportunities strategically, but random client decisions undermine hiring stability, producing inconsistent outcomes.

Patterns of inequality closely follow these dynamics. The broad participation of Rand-F + Rand-C ensures equitable distribution, while selective client rejection in Rand-F + LLM-C concentrates work among the few freelancers whose random bids happen to align with client preferences (Gini = 0.45). LLM-F + LLM-C configurations produce moderate inequality (Gini = 0.29–0.50), reflecting selective participation without extreme concentration. Taken together, these results suggest that effective marketplaces require either adaptive coordination of selectivity, as in LLM-F + LLM-C with reflections, or coordinated non-selectivity, as in Rand-F + Rand-C. Hybrids fail precisely because mismatched cognitive approaches generate coordination breakdowns. Figure 1 summarizes these selectivity-driven trade-offs with 95% confidence intervals.

## 4.3 Reflection Mechanism Ablation Study

Comparing LLM-F + LLM-C configurations with and without reflections highlights the impact of strategic adaptation. With reflections, fill rates rose from 43.5% to 69.3% and freelancer hiring rates from 63.5% to 84.6%. Inequality also dropped (Gini $0.50 \rightarrow 0.29$), and participation nearly doubled ($4.1\% \rightarrow 8.9\%$).

Interestingly, agents without reflections showed slightly higher bid efficiency (32.2% vs. 25.1%). This is because they bid less often and they are stricter ($\approx 1.3$ bids/job vs. 2.9), so a greater share of their limited bids succeeded. Reflections made agents more exploratory and less selective, lowering per-bid conversion but improving market-wide match quality and participation.

These findings demonstrate that reflection mechanisms fundamentally alter agent behavior from reactive to adaptive, enabling strategic learning that improves overall market functioning while revealing important trade-offs between exploration and exploitation in bidding strategies. The ablation study illustrates the framework's capability to isolate and measure the impact of specific cognitive mechanisms on economic outcomes.

## 4.4 Key Behavioral Mechanisms in the LLM-F + LLM-C Configuration

Having established comparative performance differences, we now examine the behavioral mechanisms driving these outcomes. Focusing on the LLM-F + LLM-C configuration with reflections, which achieved the highest strategic efficiency, we analyze four key dimensions of agent behavior.

First, rejection reasoning shows systematic skill- and fit-centric patterns on both sides. LLM clients mainly reject on skill mismatch and project alignment, while freelancers decline to bid when projects fall outside core competencies or budgets; rate sensitivity is secondary. These patterns explain why selective participation yields higher bid efficiency (see Appendix C.1; Figure 2).

Second, reputation mobility is evident within the 100-round horizon: 37% of freelancers advance beyond "New," forming a progression ladder (New→Established→Expert→Elite). Most move into the Established tier, with few reaching Expert or Elite. This suggests selective strategies generate cumulative advantages while avoiding extreme concentration (see Appendix C.2; Figure 3).

Third, category dynamics reveal significant variation in both fill rates and competition intensity across skill domains. Some categories like Data Science demonstrate high competition (approximately 10 bids per job) with strong fill rates, while others like Engineering & Architecture show more moderate competition levels (around 3 bids per job) with different completion patterns (see Appendix D; Figure 4).

Full analyses, figures, and statistical details appear in Appendices C.1–D, including rejection taxonomies, reputation trajectories, category-level performance metrics, and inference procedures.

# 5 Discussion and Conclusions

This research presents a systematic framework for investigating emergent economic behavior in LLM agent populations through controlled marketplace simulations. Our comparative experiments illuminate how cognitive architectures shape economic behavior, offering insights beyond traditional agent-based models.

## 5.1 Key Findings

**Cognitive architecture shapes market behavior.** Performance differences across configurations show that reasoning creates qualitatively distinct behaviors rather than simply optimizing existing strategies. LLM agents exhibit selective bidding, skill-based rejection reasoning, and strategic decision-making—supporting their characterization as bounded policy approximators that generate novel patterns rather than enhanced rule-based agents.

**Selectivity alignment drives efficiency.** Successful marketplace functioning depends on coordination between participants' selectivity. Aligned strategies—both sides selective (LLM-F + LLM-C) or both sides non-selective (Rand-F + Rand-C)—produce efficient outcomes, while mismatched

selectivity in hybrids leads to coordination failures. Reflection acts as a key adaptation tool, enabling dynamic adjustment based on feedback and improving collective outcomes through higher fill rates and reduced inequality despite lower per-bid efficiency.

**Multi-dimensional evaluation reveals trade-offs.** No single metric captures performance, as cognitive approaches optimize different objectives. Volume-maximizing systems achieve high throughput and equity, while quality-focused systems balance selectivity with coordination. This demonstrates a tension between volume and match quality that may generalize beyond synthetic environments, suggesting different market designs favor different cognitive approaches depending on priorities.

### 5.2 Methodological Contributions

The framework addresses limitations in traditional agent-based modeling through: (1) transparent natural language reasoning enabling inspection of logic, (2) emergent strategies from interactions rather than fixed rules, (3) longitudinal tracking of individual and market-level dynamics, and (4) systematic comparative experiments across cognitive architectures. By capturing both aggregate outcomes and reasoning traces, LLM-based simulations complement empirical work and theory in understanding how cognitive factors shape economic behavior.

### 5.3 Implications and Future Directions

These findings have broader implications for understanding AI adoption in economic systems. Asymmetric configurations show that reasoning must be balanced across participants for optimal outcomes—when only one side uses strategic reasoning (Rand-F + LLM-C), severe inefficiencies emerge, suggesting cognitive mismatches cause coordination failures. This insight suggests AI adoption in real markets could create temporary disruptions before reaching new equilibria.

Future research directions include: (1) baseline comparisons with rule-based agents and ablation studies of reputation and reflection, (2) dynamic conditions with agent entry/exit and evolving skill demands, (3) cross-model validation using different LLMs to establish generalizability, (4) larger samples for detecting subtle differences, and (5) integration with human data to validate the authenticity of observed patterns.

By showing how LLM agents can serve as controllable, transparent proxies for studying economic behavior, this work establishes a foundation for systematic investigation of market phenomena difficult to study through traditional empirical or theoretical approaches.

## 6    Limitations and Ethics

**Simulation Constraints:** Key limitations include fixed populations, discrete time rounds, simplified skill representation, and LLM-specific reasoning patterns. The framework cannot study market entry/exit dynamics, network effects, or authentic human psychological factors. Our 100-round timeframe limits observation of longer-term institutional development.

**Statistical Limitations:** Our multi-run experimental design (20 runs per configuration) enables statistical significance testing and confidence interval calculation with reasonable power for detecting the observed behavioral differences between agent reasoning approaches. However, multiple comparisons across metrics and configurations increase the risk of Type I errors. While we did not apply formal corrections given the exploratory nature of this work and the large observed effect sizes, readers should interpret statistical significance claims accordingly.

**Agent Characterization:** Our framework treats LLM agents as bounded policy approximators, enabling systematic study of their strategic behaviors through controlled experimentation. The configuration design (LLM-F + LLM-C, LLM-F + Rand-C, Rand-F + LLM-C, Rand-F + Rand-C) provides initial baseline comparisons, demonstrating how reasoning capabilities affect market outcomes across different agent combinations. Random baseline parameters (5% bid probability, 50% acceptance rate) were chosen to establish a clear noise floor but were not varied systematically.

**Research Scope** This work is designed exclusively for academic research into AI agent behavior and economic pattern formation. The simulation uses anonymized identifiers to prevent demographic

bias. Findings highlight AI agent capabilities and limitations rather than real human behavior, and should not be interpreted as assessments of existing platforms or used for policy recommendations.

**Ethical Considerations:** Because the framework produces synthetic market behaviors, there is a risk of overgeneralization to real labor markets or of misusing results to justify interventions in human employment systems. To mitigate this, we (i) emphasize that the findings are illustrative and not prescriptive, (ii) exclude demographic variables to avoid reinforcing stereotypes or bias, and (iii) release the framework openly for transparency and reproducibility, enabling scrutiny by the broader research community.

**Broader Impacts:** This work serves as a testbed for economic behavior modeling in controlled conditions, advancing simulation methodologies and enabling systematic investigation of market phenomena. Important risks include potential overgeneralization to real-world systems and misinterpretation of synthetic behaviors as reflecting human decision-making. Our findings should not be interpreted as prescriptive recommendations for existing economic systems.

# 7 Reproducibility

The complete codebase and dataset are publicly available as open-source software at `https://github.com/upwork/simploy`. The repository includes source code, configuration files, test suites, documentation and analysis scripts to enable full reproduction of our experimental results.

For complete experimental parameters, technical implementation details, and step-by-step reproduction instructions, see `REPRODUCIBILITY.md` in the repository root directory.

# 8 AI Agent Setup

GPT-5 was used within ChatGPT for brainstorming, planning, and reviewing the paper. To verify references and minimize hallucinations, we employed GPT-5 with the web search capability enabled.

Claude 4.5 Sonnet was used within Cursor (Agent Mode) for code implementation and writing the manuscript.

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

## A    Technical Implementation Details

### A.1    Agent Architecture and Implementation

#### A.1.1    Agent Types and Reasoning Models

The framework implements multiple agent types enabling controlled behavioral comparisons:

- **LLM Agents**: Use structured reasoning prompts incorporating persona, track record, strategic insights, and reputation context to make contextual decisions
- **Random Agents**: Make probabilistic decisions without reasoning (5% bid probability, uniform bid values between 50-150% of job budget)

### A.1.2 Prompt Engineering and Decision Architecture

Agent behavior emerges from multi-stage prompt engineering. Freelancer decision prompts integrate: (1) persona elements (personality, background, motivation), (2) performance history (bid acceptance rates, job completion rates, track record), (3) strategic context (market observations, reputation standing), and (4) current constraints (remaining bids, workload capacity). All decisions return structured JSON responses with reasoning, decision, and communication elements.

**Example Freelancer Decision Prompt:**

> *You are Freelancer A, a freelancer.*
>
> *YOUR PROFILE: You are Creative thinker with a strong attention to detail, enjoys collaborating with clients and 5 years of experience in various design projects for startups and established brands. Your motivation is Passionate about visual storytelling and creating impactful designs. Your skills include graphic design, branding, illustration. You charge a minimum of $40/hour and prefer medium length projects.*
>
> *YOUR TRACK RECORD: You've submitted 12 bids and been hired 3 times. Success rate: 25% (3/12) Recent projects show strong performance in Design & Creative category.*
>
> *STRATEGIC INSIGHTS: Focus on projects matching your design expertise. Budget alignment important for sustainable work. REPUTATION: New (100% success rate, $960 earned) Top skills: Design & Creative: 85.0% Market position: just starting to build your reputation*
>
> *BIDDING CONSTRAINTS: You have 2 bid(s) left this round.*
>
> *JOB OPPORTUNITY: Title: Logo Design for Tech Startup, Company: TechCorp B, Category: Design & Creative, Description: Create modern logo reflecting innovation and reliability for B2B software company, Skills Required: graphic design, branding, Budget: Fixed - $800, Timeline: 1-2 weeks*
>
> *DECISION: Should you bid on this job? Consider your skills, the budget, and whether this could be a good opportunity.*
>
> *Return a JSON object with these exact fields: - decision: "yes" or "no" - reasoning: short explanation for your decision - message: brief pitch to the client if bidding (required if decision is "yes")*

Client decision prompts similarly integrate company profiles (size, budget philosophy, hiring style), job requirements, and bid evaluations. For example:

> *You are the hiring manager at TechFlow Solutions, a medium company.*
>
> *ABOUT YOUR COMPANY: We're a growing tech consultancy specializing in digital transformation projects for mid-market clients. You have a balanced approach to budgets and a collaborative hiring style.*
>
> *YOUR JOB POSTING: Title: Data Analysis Dashboard Company: TechFlow Solutions Category: Data Science Description: Build an interactive dashboard for client KPI tracking Required Skills: Python, Tableau, SQL Budget: fixed - $2500 Timeline: medium*
>
> *You've received 3 bids to review:*
>
> *Bid #1: - Freelancer: DataViz_Expert - Skills: Python, Tableau, SQL, JavaScript - Track record: 85% success rate - Message: "I have extensive experience building interactive dashboards for KPI tracking..." ...*
>
> *DECISION: Review the bids and decide which freelancer would be the best fit for your company and project.*

The reflection system enables strategic adaptation through periodic self-assessment prompts that synthesize recent performance patterns and market observations into actionable strategy adjustments.

### A.1.3 Bias Mitigation and Persona Generation

Agent characteristics are generated using carefully designed prompts that ensure balanced representation while avoiding demographic stereotyping. Freelancer agents are assigned anonymous identifiers (Freelancer A, B, C, etc.) and generic regional labels (Region 1, 2, 3, etc.) rather than specific demographic attributes. Client companies use neutral business identifiers ("Company A", "TechCorp B") avoiding cultural associations. All agent traits including skills, pricing strategies, and behavioral patterns are randomized across the population.

# B    Agent-Level Supplementary Materials

In this section, we provide detailed information on agent behaviors, decision-making processes, persona generation, and individual-level learning mechanisms that inform agent reasoning capabilities.

## B.1    Detailed Persona Analysis

### B.1.1    Freelancer Persona Examples

The framework employs diverse freelancer personas with anonymized identifiers to avoid demographic bias. Representative examples from our illustrative experiments include:

**Freelancer A (Design & Creative):** Anonymous freelancer specializing in graphic design, branding, and illustration, minimum rate $40/hr. Personality: "Creative thinker with a strong attention to detail, enjoys collaborating with clients." Motivation: "Passionate about visual storytelling and creating impactful designs." Background: "5 years of experience in various design projects for startups and established brands."

**Freelancer B (Data Science & Analytics):** Anonymous freelancer with skills in data analysis, machine learning, and data visualization, minimum rate $75/hr. Personality: "Analytical and methodical, thrives on solving complex problems and enjoys working independently." Motivation: "Driven by the desire to derive insights from data that can influence business decisions." Background: "7 years of experience in data analytics roles within tech companies."

**Freelancer I (Translation):** Anonymous translator specializing in Spanish translation, proofreading, and editing, minimum rate $30/hr. Personality: "Meticulous and culturally aware, enjoys bridging communication gaps through language." Motivation: "Passionate about making information accessible to diverse audiences." Background: "5 years of experience in translation services for various clients and projects."

**Note on Demographic Bias Prevention:** All freelancer personas use completely anonymous identifiers with no geographic, cultural, or demographic information included in the framework. Location data has been entirely removed from persona generation, agent decision-making, and client hiring processes to eliminate potential geographic bias and ensure research focuses purely on skill-based market dynamics.

### B.1.2    Client Persona Examples

The framework employs client agents representing diverse business categories and hiring philosophies:

**Admin Solutions Inc A:** Startup company with cost-conscious budget philosophy and quick decision-making hiring style. Background: "A newly established company specializing in administrative support services for small businesses. Focuses on providing streamlined solutions at a low cost to help other startups optimize their operations." Demonstrates how startup clients prioritize cost-effective solutions and rapid decision-making.

**Sales Dynamics Corp B:** Medium-sized firm with value-focused budget philosophy and thorough evaluation hiring style. Background: "A mid-sized firm dedicated to sales consultancy and strategy development. Emphasizes building long-term partnerships with clients while delivering data-driven solutions to enhance sales performance." Shows how established companies balance quality with strategic evaluation processes.

**Design Innovations LLC C:** Small boutique agency with premium-focused budget philosophy and relationship-focused hiring style. Background: "A boutique design agency that specializes in high-end branding and visual identity projects. Known for its artistic approach and attention to detail, the company aims to create lasting impressions for its clients." Illustrates how specialized agencies prioritize quality and relationships over cost considerations.

## B.2 Reputation System Details

### B.2.1 Tier Progression Criteria

The four-tier reputation system operates with performance-based advancement criteria for both agent types:

**Freelancer Tiers:**

- **New:** < 3 jobs hired
- **Established:** 3-6 jobs hired
- **Expert:** 7-14 jobs hired
- **Elite:** 15+ jobs hired

**Client Tiers:**

- **New:** 0-4 jobs posted
- **Established:** 5-19 jobs posted with $\geq$60% hire success rate (jobs filled / jobs posted)
- **Expert:** 20-49 jobs posted with $\geq$75% hire success rate
- **Elite:** 50+ jobs posted with $\geq$85% hire success rate

# C  Market-Level and System Supplementary Materials

In this section, we provide supplementary materials on market dynamics and system-level dysfunction patterns.

## C.1  Rejection and Hiring Analysis

### C.1.1  Rejection Pattern Classification

We systematically analyzed rejection feedback using keyword-based classification to categorize rejection reasons into five thematic areas: skill mismatch (keywords: skill, experience, expertise, qualifications, background), competition (better, stronger, more, competitive, other), fit (fit, match, align, suitable, appropriate), rate/budget (rate, budget, cost, price, expensive), and communication (message, communication, understanding, proposal, pitch). Each rejection reason was examined for the presence of these keywords, with some reasons matching multiple categories. For this analysis, we sampled a run from the LLM-F + LLM-C configuration to examine detailed decision-making patterns.

To understand how LLM agents make strategic decisions, we analyzed rejection patterns when agents use reasoning capabilities.

Figure 2 shows the systematic rejection patterns for both LLM clients and freelancers.

**LLM client rejection analysis:** When clients are LLM agents, they provide detailed rejection reasoning with systematic evaluation patterns across five dimensions. Skill mismatch and project fit dominate rejection decisions, while rate/budget concerns are less common, indicating clients prioritize capability alignment over cost considerations. Examples of LLM client reasoning include: "Neither freelancer has the required experience in English to Spanish translation, as both have skills focused on Spanish..." (skill mismatch), "while you have strong technical skills, your proposal lacked specificity regarding how your experience directly aligns with our unique project goals" (project fit), and "None of the bids match the required skills for the Legal Design Project Specialist position. All bidders have experience in other areas..." (skill-job alignment).

**LLM freelancer decision analysis:** When freelancers are LLM agents, they provide detailed reasoning when choosing not to bid on job opportunities, with strategic rejections across systematic dimensions. Skill mismatch and project fit are the primary factors, while rate/budget considerations show notable economic awareness. Competition factors are minimal, suggesting freelancers focus on capability alignment rather than competitive dynamics. Examples of LLM freelancer reasoning include: "The job requires IT and networking skills that do not align with my expertise in graphic

design, UI/UX design, and branding" (skill mismatch), "The job opportunity is focused on IT & Networking, which does not align with my core skills in graphic design" (project fit), "The job opportunity requires translation skills which do not align with my expertise in graphic design and UI/UX. Additionally, the budget may not justify the time investment" (rate/budget consideration), and "This job opportunity focuses on software development and localization, which are outside my core skills in graphic design" (strategic specialization). This demonstrates how LLM agents engage in multi-factor strategic decision-making based on skill alignment, project requirements, and economic considerations.

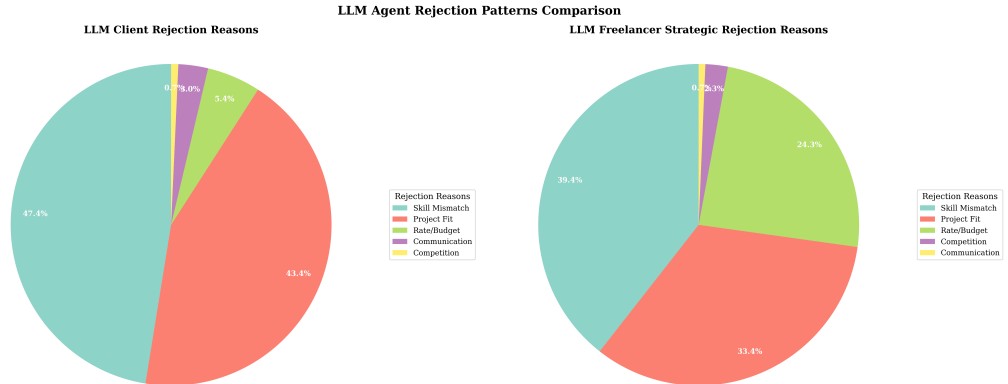

Figure 2: LLM agent rejection patterns comparison showing systematic decision-making by both clients and freelancers. Both agent types prioritize skill alignment and project fit over cost considerations, with clients showing slightly higher emphasis on skill mismatch while freelancers demonstrate greater budget awareness.

## C.2   Reputation System Analysis and Impact

Our analysis of the LLM-F + LLM-C configuration reveals significant freelancer progression through the reputation tiers over the 100-round simulation, as shown in Figure 3. Of the 200 freelancers, 37% (74 freelancers) advanced beyond the "New" tier, demonstrating meaningful skill development and performance differentiation.

It is important to note that the 100-round simulation period constrains progression opportunities, as many freelancers may not have sufficient time to accumulate the jobs required for higher tiers. The advancement rates observed represent progression within this limited timeframe and would likely increase in longer simulations.

An example of a reputation-conscious hiring decision is *"Freelancer E's skills in digital marketing and social media management align well with our requirements. Their eagerness to build a reputation suggests they will be dedicated and motivated to produce high-quality results."*

# D   Category-Based Market Analysis

This section analyzes fill rates and competition levels across skill categories in the LLM-F + LLM-C configuration to understand how market dynamics vary by domain.

Competition intensity, measured as the average number of bids per job, varies widely across categories, as illustrated in Figure 4. Data Science jobs attract approximately 10 bids per job, representing high-intensity competition, while Engineering & Architecture sees around 3 bids per job, reflecting more moderate competition levels. Design and Translation categories show intermediate competition levels with consistent bidder participation.

Fill rates also demonstrate significant variation across skill domains. Some categories achieve higher job completion rates, indicating better supply-demand matching, while others show lower fill rates despite varying levels of competition. This suggests that competition intensity and job completion success operate through different mechanisms across skill categories.

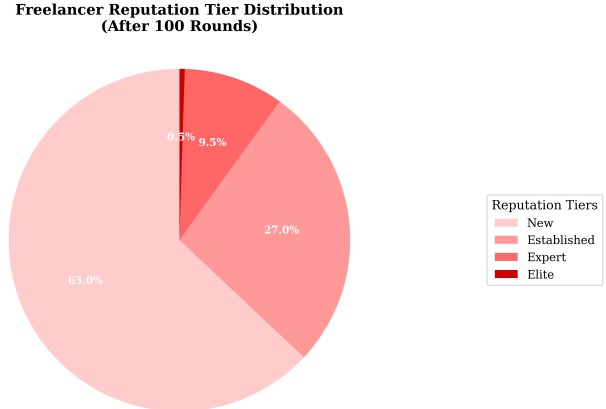

Figure 3: Freelancer reputation tier distribution after 100 rounds showing natural progression hierarchy. Most freelancers (63%) remain in the "New" tier, while 27% achieve "Established" status, 9.5% reach "Expert" level, and 0.5% (1 freelancer) attain "Elite" status. This distribution reflects realistic market dynamics where advancement requires sustained performance.

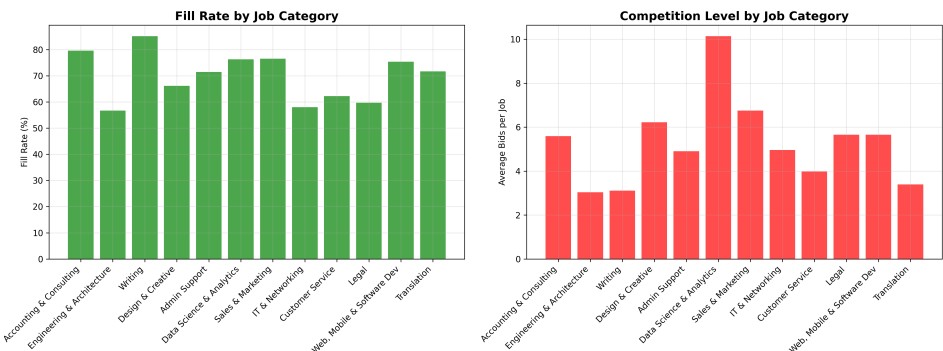

Figure 4: Fill rates and competition intensity across skill categories in the LLM-F + LLM-C configuration. Data Science demonstrates the highest competition intensity with 10 bids per job, while Engineering & Architecture shows more moderate competition. Fill rates vary significantly across categories, with some domains achieving higher job completion rates than others.

The category-level analysis reveals that market efficiency depends not only on overall agent reasoning capabilities but also on domain-specific factors that influence how freelancers and clients interact within particular skill areas.

## E  Statistical Methods

Confidence intervals (95%) were calculated using t-distribution with sample standard error, appropriate for the sample sizes (20 runs per configuration). Statistical significance tests employed independent t-tests for comparing metric values between configurations. P-values below 0.05 were considered statistically significant.

## F  Implementation Details

The simulation framework is implemented in Python 3.9+ with NumPy 1.21+, SciPy 1.7+, Matplotlib 3.4+, and OpenAI GPT API.

# G    Computational Resources

All experiments were conducted on a AWS EC2 instance with 48 CPU cores and 373 GiB of RAM. Execution times varied significantly by configuration due to the computational requirements of different agent types. The LLM-F + LLM-C configuration required approximately 50 minutes per run using 30 concurrent workers, with the primary bottleneck being network latency for API calls rather than local computational resources. Configurations involving only random agents (Rand-F + Rand-C, LLM-F + Rand-C, Rand-F + LLM-C) completed substantially faster as they do not or partially require external API calls.

## Agents4Science AI Involvement Checklist

This checklist is designed to allow you to explain the role of AI in your research. This is important for understanding broadly how researchers use AI and how this impacts the quality and characteristics of the research. **Do not remove the checklist! Papers not including the checklist will be desk rejected.** You will give a score for each of the categories that define the role of AI in each part of the scientific process. The scores are as follows:

**Human-generated**: Humans generated 95% or more of the research, with AI being of minimal involvement. **Mostly human, assisted by AI**: The research was a collaboration between humans and AI models, but humans produced the majority (>50%) of the research. **Mostly AI, assisted by human**: The research task was a collaboration between humans and AI models, but AI produced the majority (>50%) of the research. **AI-generated**: AI performed over 95% of the research. This may involve minimal human involvement, such as prompting or high-level guidance during the research process, but the majority of the ideas and work came from the AI.

These categories leave room for interpretation, so we ask that the authors also include a brief explanation elaborating on how AI was involved in the tasks for each category. Please keep your explanation to less than 150 words.

1. **Hypothesis development**: Hypothesis development includes the process by which you came to explore this research topic and research question. This can involve the background research performed by either researchers or by AI. This can also involve whether the idea was proposed by researchers or by AI. 2. **Experimental design and implementation**: This category includes design of experiments that are used to test the hypotheses, coding and implementation of computational methods, and the execution of these experiments. 3. **Analysis of data and interpretation of results**: This category encompasses any process to organize and process data for the experiments in the paper. It also includes interpretations of the results of the study. 4. **Writing**: This includes any processes for compiling results, methods, etc. into the final paper form. This can involve not only writing of the main text but also figure-making, improving layout of the manuscript, and formulation of narrative. 5. **Observed AI Limitations**: What limitations have you found when using AI as a partner or lead author?


